# Narrative Review and Analysis of the Use of “Lifestyle” in Health Psychology

**DOI:** 10.3390/ijerph20054427

**Published:** 2023-03-01

**Authors:** Francesca Brivio, Anna Viganò, Annalisa Paterna, Nicola Palena, Andrea Greco

**Affiliations:** Department of Human and Social Sciences, University of Bergamo, 24129 Bergamo, Italy

**Keywords:** lifestyle, narrative review, health psychology, health practice

## Abstract

Lifestyle is a complex and often generic concept that has been used and defined in different ways in scientific research. Currently, there is no single definition of lifestyle, and various fields of knowledge have developed theories and research variables that are also distant from each other. This paper is a narrative review of the literature and an analysis of the concept of lifestyle and its relationship to health. This contribution aims to shed light on the lifestyle construct in health psychology. In particular, the first part of this manuscript reexamines the main definitions of lifestyle in the psychological and sociological fields through three perspectives: internal, external, and temporal. The main components that characterise lifestyle are highlighted. The second part of this paper explores the main concepts of lifestyle in health, underlining their strengths and weaknesses, and proposes an alternative definition of a healthy lifestyle, which integrates the individual dimensions with the social and cycle dimensions of life. In conclusion, a brief indication of a research agenda is presented.

## 1. Introduction

Lifestyle is a complex and often generic concept that has been used and defined in different ways in scientific research. Currently, there is no single definition of lifestyle and the various fields of knowledge have developed theories and research variables that are also distant from each other. The term appeared for the first time in the mid-eighteenth century when the French naturalist and writer Georges Louis de Buffon (1707–1788) stated, “The style is the man himself” [1]. In the past, 150 years ago, the English philosopher and author Robert Burton observed, “It is most true, stylus virum arguit-our style betrays us” [2] (p. 122). Lifestyle was associated with these first references as a “manner of expression” [3].

Lifestyle has been used and explored as a construct within sociological currents, considering, for example, the theories of Weber and Bourdieu [4,5] or recently [6] in which it is viewed as an expression of social class. While the psychological perspective defines and analyses lifestyle on the level of thought or the level of action. In particular, lifestyle has been addressed in the context of consumer psychology [7], psychology of values [8,9,10,11], and individual psychology [12,13].

In health psychology, the use of this concept has spread widely, especially in the preventive medicine sector, despite a definition that is not always precise and unambiguous. Indeed, it is often confused and assimilated to the health behaviours that have been defined as “…overt behavioural patterns, actions and habits that relate to health maintenance, to health restoration and health improvement” [14] (p. 3). However, there are differences between these constructs, which we are going to clarify in this article.

The difference between a healthy lifestyle and health behaviours appears to be subtle and not always clear.

Within the sociomedical discourse, there is a frequent tendency to use the concept of lifestyle as individual behavioural patterns, which influence the status of the disease [15] and can be modified with targeted educational campaigns [16,17]. Research in this area investigates lifestyle in terms of the presence or absence and frequency of “unhealthy” behaviours (smoking, alcohol, diet, and physical activity) [18,19,20]. The exclusive focus on risk behaviours reflects the typical tendency of the risk society [21] to identify the pursuit of health to avoid risk factors. In this scenario, the subject is responsible and the protagonist of their own health choices [22]. Certainly, lifestyle is a historic research topic in the field of medicine, epidemiology, and health psychology. This is justified by the modern condition of epidemiological scenarios. Indeed, the World Health Organization (WHO) has been highlighting for decades how the fight against chronic diseases represents the challenge of the 21st century; noncommunicable diseases (NCDs), such as cardiovascular disease, cancer, diabetes, and chronic respiratory diseases, represent the leading cause of death worldwide and are responsible for 70% of deaths globally [23]. The risk factors on which the WHO focuses are precisely lifestyles, which can be traced in the following behaviours: use of tobacco, unhealthy diet, lack of physical activity, and excessive alcohol consumption, which in turn lead to overweight and obesity, increased blood pressure, and increased cholesterol, all risk factors for the onset of a disease. Furthermore, research has highlighted significant correlations between personality characteristics and risk behaviours or lifestyles, which stand as predictors of adverse health outcomes [24,25,26]. Matarazzo [27] coined the concept of pathogenic behaviours to encapsulate the notion of risky lifestyles.

Focusing on the preventive and risk avoidance perspective does not seem to be sufficient for implementing health promotion action and intervention strategies, especially when considering health as a complex topic, such as a state of complete bio–psycho–social wellbeing rather than just the absence of disease, as stated in the historical conception of WHO [28], as. Furthermore, recent revision define health as the ability to adapt and cope autonomously with life’s ever changing physical, emotional, and social challenges [29].

According to the WHO definition, the close relationship between health and wellbeing is clear. However, it is important to note that wellbeing is defined here as “An umbrella term for different valuations that people make regarding their lives, the events happening to them, their bodies and minds, and the circumstances in which they live” [30] (p. 400)

It is also essential to consider that the COVID-19 emergency has led to a significant change in people’s daily life and lifestyle, with important social, work, and educational implications. New vulnerabilities and the worsening of health inequalities have emerged, strongly affecting the individual and community on several fronts, highlighting, on the one hand, individual and social protective factors and, on the other, factors of vulnerability.

Thus, it is critical to investigate how lifestyle is conceptualised in the field of health, particularly health psychology, as a fundamental step to build more effective theoretical and explanatory models, which would be the basis of health promotion interventions. To reach a critical reworking of a healthy lifestyle concept, we believe it is essential to attempt a multidisciplinary analysis, primarily sociological and psychological perspectives, on the theories and definitions of lifestyle to identify the salient elements that characterise this construct. Currently, no review has debated the multidisciplinary conceptualisations of lifestyle.

This contribution aims to shed light on the lifestyle construct in health psychology, trying to conceptually redefine it to propose a definition that would allow to combine individual, social, and life span dimensions. This reflection is developed within a salutogenic and ecosystemic perspective of health [31,32]; furthermore, the bio–psycho–social wellbeing dimensions are considered within the life cycle and a historical framework as well. Therefore, health is constructed within a continuous exchange and repositioning between subject and environment [32,33,34,35,36]. To pursue the purpose of this paper, we propose a narrative review of the psychological and sociological fields’ main definitions and conceptualisations of lifestyle. We provide particular importance to theories that relate the subjective–psychological dimensions with the social and environmental. We chose a narrative review because it allows for acquiring a greater understanding of a given topic, why it was studied it in a particular way, and the interpretations given [37].

The first part of the paper presents the various definitions, results of studies, and theories of different disciplines, trying to reconsider them along three perspectives: internal, external, and temporal. The intent is to understand the main components that characterise lifestyle. The second part of the paper explores the main conceptualisations of lifestyle in health, underlining their strengths and weaknesses. In the conclusions, a definition of lifestyle in health is proposed as a starting point for trying to advance future theoretical and research perspectives.

## 2. Materials and Methods

The narrative review plays an important role in expanding our understanding not only of the topic of lifestyle but also of the reasons why it has been studied in a particular way, the definitions and interpretations that have been variously made with respect to what we know about it, and the nature of the knowledge that informs or may inform research and intervention practice [37].

We conducted a keyword search-based literature review using Scopus, PsycInfo, Web of Science, and Google Scholar search for studies with titles or abstracts containing “lifestyle” OR “life style” AND (“definition”) AND (“health” OR “healthy”). We included English-language-based, international peer-reviewed articles (e.g., reviews), online reports, and electronic books.

We applied a snowballing search methodology using the references cited in the articles identified in the literature search. Each identified item was assessed for relevance by a member of the study team, and we included articles that examined lifestyle, as well as the way it was expressed. This review is not comprehensive but is intended to summarise literature on lifestyle definition in psychological and sociological area. We have not indicated time limits, through January 2022.

## 3. Results

### 3.1. The Concepts of Lifestyle

Over time, several reviews have been carried out on the concept of lifestyle, among which the main ones are mentioned [3,7,38,39,40], which allowed us to further investigate this construct exploring the different facets. Table 1 displays a collection of the most recognised and cited definitions in the literature, from the first formulations to the most recent ones, considering the ambitions of psychological and sociological research. Starting from the categorisation of the content of the definitions, three interpretative keys were proposed: internal, external, and temporal (see Table 1).

Internal dimension: Lifestyle as a synonym for personality style, an expression of cognitive styles, or a set of attitudes, interests, and values. The focus is placed on the subject and on the internal processes that guide behaviour and action;External dimension: lifestyle as an expression of the individual’s status and social position within a given context or as an expression of behavioural patterns;Temporal dimension: lifestyle as a stable dimension that is expressed within daily practices; this dimension is found transversally in some sociological and psychological perspectives.

#### 3.1.1. Internal Dimension

The lifestyle concept, especially in the psychological field, has followed two main perspectives and research: a first perspective, the initial one, identified lifestyle as a personality trait of the subject, as an expression of human creativity. In this unique psychic imprint that characterises each individual, the traits of behaviour, thoughts, opinions, emotions, and feelings converge, resulting from a compromise between individual needs and social demands [3,12].

Adler [12] was one of the first researchers that used the lifestyle construct. In “psychological means and ways for the investigation of the life style”, Adler [12] pointed out that poets have always described lifestyles, although what they did was not formulated in this way. “Our knowledge of the individual is very old. To name only a few instances, the historical and personality descriptions of the ancient peoples, the Bible, Homer, Plutarch, all the Greek and Roman poets, sagas, fairy tales, and myths, show a brilliant understanding of personality. Until recent times it was chiefly the poets who best succeeded in getting the clue to a person’s life style. Their ability to show the individual living, acting, and dying as an indivisible whole in closest context with the tasks of his sphere of life rouses our admiration for their work to the highest degree” [12] (pp. 32–33). According to the author, individuals adopted a particular lifestyle to overcome their inferiority and their social interaction problems. The lifestyle would have been associated with the fundamentally personal character that was defined in childhood and governed reactions and behaviour [45]. Each individual built their vision of the world in the first 4 or 5 years of life. Adler considered the person as a whole, and the set of values and guiding principles represent the lifestyle. In this first theory, lifestyle was defined without exploring in depth the process that led to its constitution. It was also unclear what were the set of underlying values and principles and how they differed among groups. The questions of measurement and how values interacted and influenced an individual’s behaviour also remained unresolved [38].

Individual psychology, introduced by Adler, is a theory of human behaviour and a therapeutic approach that encourages individuals to make positive contributions to society and achieve personal happiness [46]. This perspective involved the lifestyle concept and its definitions as the more mature and evolved organisation of the individual personality, which emerges in adulthood. During development, an individual’s way of thinking, acting, and perceiving evolved, resulting in a specific modus vivendi or lifestyle [13,47].

Allport [13] described individual lifestyle as functionally autonomous and as the highest level of organisation of a personality, “the complex propriate organisation that determines the ‘total posture’ of a mature life-system.” Lifestyle “evolves gradually in the course of life, and day by day guides and unifies all, or at least many, of a person’s transactions with life” [13] (p. 237). Each individual tends to establish a unique lifestyle that characterises every action and thought and distinguishes them from all others.

Coleman stated that “the individual’s pattern of assumptions leads to consistent ways of perceiving, thinking, and acting-to a characteristic modus operandi or life style” [41] (p. 63). “Each individual tends to establish a unique relatively consistent life style. He has a characteristic way of going, thinking, reacting, and growing that tends to distinguish him from everyone else. He puts his personal stamp on every role he plays and every situation he encounters ... consistent with his self-concept” [3,47] (p. 69). The coherence of a lifestyle given by a continuous pattern of assumptions and attitudes makes the individual’s behaviour somewhat predictable.

In contrast with the first perspective presented above, the second line of psychological research identifies lifestyle considering values, attitudes, and interests. In the 1960s, Rokeach linked the concept of lifestyle with the value system; according to this author, each person had a few hierarchically ordered values. Values were differentiated into terminal values, when they referred to an individual or collective existence, and instrumental values, namely, behavioural models, ways of acting, and being. Values and attitudes converged in a hierarchical system that tended to be stable over time [48].

The VALS (values and lifestyle) perspective proposed by Mitchell [8,9,49,50], starting from Maslow’s studies and theories [51,52,53], was different. Not far from Adler’s definition, Mitchell defined lifestyle as follows: “we started from the premise that an individual’s array of inner values would create specific matching patterns of outer behavior—that is, of lifestyle” [9] (p. 23). According to this perspective, the population would have been divided based on a hierarchy of needs: physiological needs, security, love, belonging, esteem, knowledge, aesthetic satisfaction, and self-realisation. Needs were organised hierarchically, from the physiological ones at the bottom to the self-fulfilling ones at the top. They were the basis of the motivations that move the subject’s actions; when the primary needs were satisfied, the other levels emerged. Once the physiological and safety needs were met, the subject could take two paths: the self-directed one or the hetero directed one; the final point of arrival was common, namely, self-realisation. Lifestyle, defined by sociodemographic traits, attitudes, behaviours, and values, determines different methods and possibilities for satisfying needs. For example, there will be those people who are more oriented toward the need for security, others focused on the need for social recognition (outer-directed), and others more focused on the need for self-gratification (inner-directed).

In the second theorisation of the same author, named VALS2 (values and lifestyle), the sociological variables took on greater importance. Different lifestyles corresponded to various identified groups, which differed in a number of variables: sociodemographic factors, attitudes, economic status, consumption patterns, and practices. The population was then segmented from the meeting point among resources (education, intelligence, health, income, and self-confidence) and how principles, social status, or actions guided the individual [8,9,50].

Various researchers have developed and analysed the study approach pioneered by Mitchell, including Shultz et al. [11], who defined lifestyle as “the orientation of self, others, and society that each individual develops and follow […] [it] reflects the values and cognitive style of individual. This orientation is derived from personal beliefs based on cultural context and the psycho-social milieu related to the stages of the individual’s life” [11] (p. 4). These studies investigated the implications on the everyday action and the subject’s choices starting from emotional investment and energies in the different fields of life, ways of managing the roles assumed, and to the amount of time invested in the different roles, which concern the various fields of life.

The way in which the concept of lifestyle has been operationalised reflects the research area in which it has developed. For example, in the 1960s and 1970s, in marketing psychology, it is treated and investigated from the perspective of activities, interests, and opinions [54,55,56,57,58]. The AIO approach (attitudes, interests, and opinions) inaugurated a way of considering lifestyle in which sociocultural variables became central; one of the research objects of this current was understanding how the sociocultural context influenced opinions and attitudes and the consumption behaviour that evolved. This research line sought to understand the link between personality traits and purchasing behaviours and considered lifestyle as a general way of living, using time, and spend money [59]; however, a solid theoretical conceptualisation seemed to be lacking [60]. Cathelat’s [61,62,63] studies were also a part of this research line in which there was an effort to keep together the individual dimension, which characterised the first current of research on the style of life, with the social one in approaching the study of lifestyle. For Cathelat, lifestyle was “a system of organisation of people and things, which takes into account their relations of strength, their dialectic, their reciprocal positioning; a system of understanding individuals in the socio-cultural context” [63] (p. 41). The author considered the following variables: sociodemographic, social styles (i.e., behaviours), and cultural flows (i.e., attitudes and opinions). Lifestyles were defined as a dynamic process through which subjects socialised and submitted to stereotypes and social values. Furthermore, they represented an expression of individual motivations and behaviours. Sociocultural change was observed by transforming and evolving the population’s attitudes toward a set of values, which were segmented into a small number of sociocultural and behavioural categories identified as mentality. The latter were configured within a map drawn by two axes, rigour–pleasure, and stability and change. Each segment of the map was referred to as a sociostyle. Lifestyle represented how individuals socialise and how culture and society changed and took on new forms through socialisation. This model emphasised how individual behaviours resulted from both static and dynamic forces at a sociocultural level.

A similar direction was taken by Stebbins, who said that “A lifestyle is a distinctive set of shared patterns of tangible behaviour that is organised around a set of coherent interests or social condition or both, that is explained and justified by a set of values, attitudes, and orientations and that, under certain conditions, becomes the basis for a separate, common social identity for its participants” [7] (p. 350). Stebbins’ definition introduced the theme of behavioural patterns influenced by personal interests, social conditions, and social identity construct. Lifestyle would have originated within everyday life and would be expressed within it (Stebbins, 1997). Lifestyles were not entirely individual but were constructed through affiliation, negotiation, and the active integration of the individual and society, which were constantly reproduced through each individual [7].

The definitions that prefer an internal interpretation of lifestyle can, therefore, be summarised as follows: lifestyle as an organisation of personality, a system of values, or a pattern of behaviour justified by values, attitudes, and orientations. These conceptualisations had greater importance to the internal dimensions of thought than the external ones, which is related to action and behaviour. Criticisms against these currents of research concern the use of personality traits as variables that are too distant to justify their direct impact on behaviour. On the other hand, the privileged reference to the context of consumption does not exhaust the field of lifestyle [64].

#### 3.1.2. External Dimension

Lifestyle represents the expression of social positioning or the set of individual and collective behavioural patterns. Researchers focus on actions and behaviours within the external dimension of lifestyle or the social structure within which the subject lives, determining action possibilities.

##### Social Positioning

Weber [65] represents one of the theorists to whom several definitions of lifestyle refer, even if he never explicitly defined this construct. He was one of the first to refer directly to the concept; indeed, in *Economy and Society* [65], the term Lebensstil, which has been translated into lifestyle, identified the social forms through which the prestige of one’s social class was expressed. The types of housing, clothing, consumption, free time, body care, and ways of speaking symbolised the different lifestyles. According to Weber, society was divided into classes and status groups; classes were identified and constituted starting from the systems of production and purchase of goods; instead, status groups identified themselves starting from the forms of consumption and the ways of life conducted (or lifestyles). Status group, therefore, resided in the sphere of honour, prestige, and the social order, while the classes were within the economic system.

Ansbacher [3] emphasised how Weber used the concept of lifestyle in collective terms to express the culture of a particular social group, which identified and differentiated the individuals who belong to it. The differences in education would have determined diversities. Other readings underlined how lifestyles were identified as patterns of individual behaviours resulting from personal choices, influenced by belonging to particular status groups, distinguished by boundaries of the social context and structures [66,67].

However, Weber [65] did not associate lifestyle with people but with the status group, thus showing that they were primarily a collective social phenomenon. Status groups were aggregates of people with similar status and school backgrounds and derived from sharing similar lifestyles. Therefore, people who wished to be part of a particular state group were required to adopt the appropriate lifestyle. These groups were stratified according to their consumption patterns. These models established differences between the groups and expressed differences that already existed [5].

The criticism that Veal [38] directed at Weber was that of not having defined the lifestyle as such, indeed, a list of practices and behavioural styles did not constitute a definition.

Bourdieu [5], who reconsidered and amplified Weber’s theories, introduced the concept of lifestyle concerning what he defined as “habitus”, the generating principle of objectively classifiable practices and classification system of these practices. “It is precisely in the relationship between these two capacities that define the habitus, the ability to produce classifiable practices and works and the ability to distinguish and evaluate these practices and products (taste) that constitutes the image of the social world, that is space of lifestyles” [5] (p. 174). Thus, lifestyles were the systematic products of habits that, perceived in reciprocal relationships, became systems of signs endowed with a social qualification. The formula at the origin of lifestyle was composed of taste and the propensity and aptitude for material and symbolic appropriation of a specific class of practices or objects. According to the author, the action of individuals depended both on external causes (the social structure that defines power relations), which was the “field”, and on internal causes (beliefs, emotions, expectations, and interests), which constituted the “habitus”. Therefore, social practices resulted from the encounter between the internalisation of social structures external to the individual’s mind (social and cultural conventions) and the individual’s inclinations, preferences, and interpretations. Thus, lifestyles were socially recognisable as differentiating social groups [5]. Finally, Bourdieu underlined the contrast between a lifestyle of “necessity” of the working classes, who “did not know how to live”, and a lifestyle as a “legitimate art of living”, which was typical of the wealthy classes.

In subsequent papers, Dean et al. [68] described lifestyle as a sociocultural phenomenon, arguing that patterns of behaviour interacted with the situational context to create a lifestyle. Cultural values and beliefs shaped behavioural practices that were constrained or encouraged by specific socioeconomic conditions.

##### Practice and Behaviour

Since the 1960s, a course of study on lifestyles has developed, focusing on acting in terms of consumption, daily activities, or behaviours.

The first trend took into consideration consumption as a starting point for defining lifestyles. Berkman and Gilson [69] defined lifestyle as a unitary set of behavioural patterns that determined consumption. In this framework, lifestyle was conceptualised in nonverbal expression that manifested itself in consumption attitudes and behaviours. Therefore, the lifestyle would have had an essential function in individual identification and with the social context of belonging through the communication of one’s status-role in daily life. Through the lifestyle, the person attributed meaning to their daily life [66,70,71].

The encounter between individual and social in the signification process of daily practices appeared central in the theories of Giddens [6], who proposed this definition of lifestyle: “A lifestyle can be defined as a more or less integrated set of practices which an individual embraces, not only because such practices fulfil utilitarian needs, but because they give material form to a particular narrative of self-identity” [6] (p. 81). “Lifestyles are routine practices, the routines incorporated into habits of dress, eating, modes of acting and favoured milieus for encountering others; but the routines followed are reflexively open to change in the light of the mobile nature of self-identity” [6] (p. 106). In this definition, lifestyle would have been expressed in daily practices, but it would have been the expression and synthesis of self-realisation processes in which the subjects were reflexively engaged in organising their everyday life within a given social–cultural context. Therefore, lifestyles would have been the result of the expression of actions, transformations of living conditions, and the product of the same conditions [6,72]. For Giddens, lifestyle was not unitary but the expression and consequence of different places and environments; through a process of “disembedding”, social relations were uprooted from the local social context to restructure indefinite space–time dimensions. This led to fragmentation and diversification, on the one hand, and a search for consistency, on the other.

Veal [38] offered a different position where through a wealthy review of lifestyle, the characterising elements of the construct present in the literature were identified, and a new definition was reached. Regarding consistency, Veal wrote: “It seems then that lifestyles consist of sets of activities and practices which: either (a) ‘fit together’ as a result of some guiding set of coherent moral or aesthetic principles; (b) ‘fit together’ but only from force of circumstance (such ad age, income, household/family situation, geography); (c) do not ‘fit together’. We may conclude therefore that, although coherence is likely to be a key variable in analysing, it is not a necessary component of the definition of lifestyle, since some lifestyles may lack coherence” [38] (p. 244). For Veal, the characteristics necessary to define what was a lifestyle were a set of day-to-day activities, the levels of individual and group analysis, and the theme of choice, although the degree of freedom of choice varied from individual to individual, from group to group, and from time to time in Western societies. In Veal’s definition, lifestyle, formed through a process of wide or limited choice, was identified as a pattern of behaviour that involved the individual and the group, which were linked to sociodemographic values and characteristics and could involve different degrees of social interaction, coherence, and recognition [38]. The author highlighted that direct contact among individuals was not a necessary prerequisite for sharing lifestyles; furthermore, when group interaction had importance in defining a distinctive lifestyle, it was possible to speak of subcultures. An important step in the analysis proposed by Veal concerned the invitation not to dwell only on the identification of groups of lifestyles but to explore the formation and adoption of lifestyles themselves. This reflection appears to be very current in the field of health psychology, where studies appear limited. Veal’s approach to lifestyles was then taken up by further elaborations, such as that of Jensen for whom “A lifestyle is a pattern of repeated acts that are both dynamic and to some degree hidden to the individual, and they involve the use of artefacts. This lifestyle is founded on beliefs about the world, and its constancy over time is led by Intentions to attain goals or sub-goals that are desired. In other words, a lifestyle is a set of habits that are directed by the same main goal” [44] (p. 225) or for Starr “the basic complementary set of the material dimensions of how people live” [73] (p. 30).

Within psychology, in Jensen’s definition, developed in the field of cognitive science, it was possible to identify four a keyword: artefacts used in everyday acts, differentiated into mental artefacts (i.e., symbols such as words or numbers that could be used in cognitive processes) and physical artefacts, such as technologies, believe, intention, and desire. According to Jensen [44], lifestyle was based on how we think we know the world. Moreover, consistency over time was driven by intentions, i.e., a desired goal and the belief that it could have been achieved with reasonable means [74,75,76,77].

The definitions that prefer an external interpretation of lifestyle can, therefore, be summarised as follows: lifestyle as an expression of social position and its manifestation in behaviour, activities, and daily practices. Unlike the previous theories presented in the “internal dimension” paragraph, the lifestyle’s origin is social, and it concerns belonging to specific social groups or “classes”.

#### 3.1.3. Temporal Dimension

The characteristic of temporality in lifestyles is presented in the different conceptualisations within two frameworks: lifestyle as a stable characteristic of the individual or the social group (see the theories of Adler [45], Rokeach [48], Weber [65] and Bourdieu [5]) or lifestyle as a set of practices and patterns of behaviour that occur in everyday life and may change or evolve in particular phases of life or due to the influence of sociocultural conditions (see Giddens and Veal [6,38]).

From a sociological perspective, for example, lifestyle was expressed in daily practices and in habits but maintained its internal consistency and stability as the basis for individual safety [6]. The identity function of lifestyles would have manifested itself in particular sensitive phases of one’s personal biography, for example, adolescence or early adulthood. The choice of a lifestyle became an expression and possibility of identifying one’s distinct group of belonging instead of an outgroup.

Similarly, also in psychology, in researching the causes of lifestyle change, psychologists look at personality, thought patterns, behaviours, traits, and responses that reflect goals, tasks, emotionality, motivation, and temperament [78,79]. Some scholars have argued that the personality stabilises at the age of 30 [80]; others have claimed that it is subject to constant changes, even during the late season adulthood [81]. In a meta-analysis, Roberts and Delvecchio [82] concluded that personality remains stable over time, even if changes could be seen in the middle and later years. Thus, the change in personality attributes would have affected lifestyles throughout life.

The similarity of the sociological and psychological perspectives characterises the temporal dimension of lifestyle. Transversal limits to the different approaches concern the origin of lifestyles and the lack of clearness on how they form and evolve. Do lifestyles change in a linear or nonlinear way (for example, as a result from individual and/or collective crises and changes)? Concerning temporality, it is not clear in the theories presented whether and how lifestyles change in the different phases of life.

### 3.2. Lifestyle in the Field of Health Psychology

Currently, in the field of health psychology, there are two main definitions of lifestyles. The first one was formulated by the WHO, for which lifestyle is defined as “patterns of (behavioural) choices from the alternatives that are available to people according to their socio-economic circumstances and the ease with which they are able to choose certain ones over others” [42]. This definition highlights that lifestyles are behavioural patterns of individual choice, influenced by the socioeconomic context in which the person lives. On the one hand, the responsibility for one’s choices is emphasised, with individual agency as the primary source of health and the prevention of pathologies; on the other, the focus is on health determinants as factors that combine themselves to define the possibilities of choice.

The second major definition of lifestyle formulated by Cockerham is “collective patterns of health-related behaviour based on choices from options available to people according to their life chances” [83] (p. 55). This definition formulated starting from the thought of Weber [84] and Bourdieu [5], to which the scholar refers directly in the model, postulated that lifestyle was comparable to a set of personal routines, which reflected belonging to certain social classes or groups in which the person was included. The set of healthy behaviours were thus grouped into lifestyles. The person coherently chose their lifestyle due to the fact of their choices and chances, which were structurally determined by socioeconomic status (SES), age, sex, race, collectivities (social networks associated with marriage, religion, politics, ideology, workplace, etc.), and living conditions. Choices and possibilities interacted with each other and influenced the formation of dispositions to act (i.e., habitus), leading to specific health-related practices (action) [5]. Both definitions have the advantage of underlining the influence of the social environment on behaviours and behavioural choices related to health, highlighting how the individual is not a monad but is inserted within a socioeconomic context that limits opportunities and personal possibilities. Therefore, these definitions align with the line of research that emphasises the importance of considering the determinants of health as factors that influence individual possibilities [85]; moreover, they refer to the wide range of social, economic, political, psychosocial and behavioural factors that directly or indirectly affect health outcomes, which in turn contribute to health inequalities [86,87,88,89,90]. Although the two main definitions focus on the influence of contextual factors in defining the individual’s possibilities of choice, the literature in this area focuses on individual behaviour at the expense of the context [91]. The main limitation of these definitions is reducing the healthy lifestyle to behavioural patterns or patterns of behavioural choice that are normatively defined and linked only to physical health; thus, little attention is given to the psychological and life cycle dimensions [92].

As Frohlich et al. [15] (p. 783, 784) put it, “…behaviours are studied independently of the social context, in isolation from other individuals, and as practices devoid of social meaning”.

The theoretical models used in this research are based on individual psychology.

Healthy lifestyles are depicted mainly as individually constructed sets of behaviours. The elements of a healthy lifestyle are described as independent of each other; the only characteristic in common is pursuing health [93].

Lifestyle, therefore, appears to be characterised in terms of behavioural models to which the subject must adhere. The subject is represented as a naive scientist, who simplistically test hypotheses, or an accountants, who evaluate the costs and benefits in behavioural change theories [94].

Intervention research on lifestyles in health psychology appears to have been dominated in history by a predominantly cognitive approach, for which it is assumed that a healthy lifestyle choice depends mainly on the subject and is influenced by a series of factors all rigorously individuals, such as self-efficacy, motivation, control and subjective beliefs [22,95,96]. Crawford [97] coined the term healthism, a form of awareness and responsibility for one’s own health and increased individual focus on prevention practices [98,99]. In this scenario, a morality of health is promoted containing specific norms and values that emphasise an individual’s obligation to worry about their health [100]—being healthy means living a balanced and controlled existence, valuing vigilance, self-control, and risk prevention.

Although important for understanding, the individual factors underlying the adoption of a healthy lifestyle and the abovementioned approaches risk being reductive concerning the complexity of the study of health. Mielewczyk and Willig [101] argue that health behaviours take on meaning only when they are considered as social practices within a specific context, the “wider social practices of which such actions form a part” [101] (p. 829). Health behaviours are deeply impregnated with broader social meanings. Practices are interconnected with social relationships [102]. As stated earlier, health and disease are intertwined in broader social, cultural, political, and historical contexts [103,104]. Lifestyles are closely linked to the habits that affect people’s daily lives. As contemporary research on intersectionality and health has shown, individuals occupy multiple social identities, or social positions, which reflect interconnected systems of power and privilege; these systems configure access to risks and resources, which ultimately shape health disparities [105,106,107]. Concerning this, Alcàntara and colleagues [86] highlight the need to adopt an evolutionary approach to the study of health disparities and how exposure to health determinants, such as marginalisation and poverty, unfold overtime on stages of development.

The development of healthy lifestyles appears to be the product of a combination of consistency and inconsistency. Healthy lifestyles are not uniformly positive or negative at different life phases and vary among sociodemographically similar people [83,108,109,110,111]. Considering health within a malaise–wellbeing continuum, it is likely to identify healthy and unhealthy behaviours within the same person. These behaviours sometimes reflect social states, such as gender, and occasionally suggest complex interactions of unmeasured social influences and human action.

As for the intervention, health promotion campaigns are often distant from the sociocultural environment of people’s lives; universal strategies such as social marketing campaigns tend to work best with people who have access to a range of social and economic resources. However, studies point out that these campaigns tend to significantly generate less improvement with low socioeconomic status (SES) or other disadvantaged groups [112,113,114]. Therefore, the overall effect could be to reinforce or exacerbate inequality in health behaviour and, hence, health outcomes, as it has been found with several tobacco control campaigns [113,114,115,116]. Implicitly, in these campaigns there is the idea that people choose the lifestyles they adopt and can engage in positive health behaviours and refrain from engaging in negative health behaviours [117]. This use of positive and negative derives from norms defined by biomedical knowledge; indeed, medicine has assumed a fundamental role in the normalisation of social life, defining healthy or unhealthy behaviours or conditions that fall on one side or the other of the confines of the constructed norm. Instead, little attention has been given to community perspectives that consider how social, cultural, and economic factors can influence people’s access to healthier lifestyles [118].

## 4. Discussion

### 4.1. Toward a Perspective of Definition and Research on Lifestyle

A healthy lifestyle has often been confused in various studies and research with healthy behaviours or avoiding risky behaviours. Although the behavioural component is important within a lifestyle, it appears to be insufficient. Currently, there is no unambiguous, shared, and recognised definition of lifestyle. Nowadays, the existing definitions lead back to patterns of behaviour or patterns of behavioural choices influenced by socioeconomic conditions [42,83]. These theories emphasise the role of health determinants [86,87,88] but limit lifestyle to the behavioural component of individual’s choice, giving little importance to the psychological, identity, and life span dimensions. On the other hand, research on lifestyle in health psychology has mainly focused on cognitive and individual factors, isolating from health practices those behaviours that take on meaning and significance only when investigated within social and cultural contexts [22,101,119]. To arrive at an alternative definition and model of a healthy lifestyle, in this narrative review, we tried to reconstruct the main models and theories of lifestyle in the psychological and sociological fields, analysing them starting from three foci: internal dimension, external dimension, and temporal dimension. This analysis made it possible to identify the main components that characterise a lifestyle.

Theories that emphasise the internal dimension focus mainly on the individual rather than on the community and are represented by two perspectives. The first current defines lifestyle as a style of personality [12,13,47], an expression of human uniqueness and creativity. These conceptualisations more directly recall the origin of the word style in the artistic field to highlight the human imprint [3]. However, such models have as their principal limit that they do not consider the social, cultural, historical, and economic context within which the lifestyle develops. Moreover, although they present a certain degree of in-depth analysis from a theoretical point of view, little attention is given to the empirical dimension [38,64]. At the same time, the second line places greater importance on values, attitudes, interests, and opinions. Still, it limits its fields of research and investigation to the level of consumption, which cannot exhaust the field of lifestyles.

Instead, in the theories that focus on the external dimension, lifestyle is expressed through behaviours that are the product of the individual’s social position within a sociocultural–economic context. The theories are mainly of sociological origin and can be differentiated based on the main focus: the social positioning for the models of Weber and Bourdieau [5,65] and the emphasis on practice and behaviour, especially related to consumer research [6,38,44]. General criticisms presented by Berzano and Genova [64] on this type of model concern the idea that there must be a social stratification at the base of a lifestyle, which is no longer sustainable for post-modernity. In addition, excessive emphasis is placed on structural factors, such as thought-generating elements and actions, underlying a lifestyle. A final critical issue outlined for some proposals concerns the excessive importance assigned to the consumption plan to the detriment of other elements.

The temporal dimension appears transversal to the different models. Issues that are still open today: How are lifestyles formed? How do they evolve over time and through the different times of life? Instead, there is agreement on considering every day as the context within which the lifestyle manifests itself [6,7,38].

The three dimensions of lifestyle (i.e., internal, external, and temporal) are considered fundamental and the main elements to outline a new definition of a healthy lifestyle. A new definition of lifestyle is therefore proposed: *lifestyle as a system of meanings, attitudes, and values within which the subject acts, which define individual and collective models of health practices within social, historical, and cultural contexts.*
Figure 1 depicts this proposal below to clarify the definition.

We analyse the different components of the proposed definition below.

### 4.2. System of Meanings, Attitudes, and Values within Which Subject Acts

The internal and external dimensions are uniting. The set of meanings, attitudes, and values represent the framework within which the subject acts. Meanings define the practice expressive content and how the subject interprets the practice [120], which may carry a different meaning within different meaning contexts [121]. By attitudes, we mean instead the “*summary evaluation of psychological object captured in such attribute dimension as good, bad, harmful-beneficial, pleasant-unpleasant and likable-dislikable*” [122] (p. 28).

Attitudes, meanings, and values precede and provide the foundation of health practices. Values represent the internalised social representations or moral beliefs that are the basis of an individual’s actions at an individual level. On the other hand, at the group level, values are scripts or cultural ideals shared by the group members [10]. Therefore, values are relatively stable beliefs about desirable acting or being, goals, and motivations that guide thinking and behaviour in everyday practices [123,124].

### 4.3. Define Individual and Collective Models of Health Practices within Social, Historical, and Cultural Contexts

Health behaviours are assumed to be constituent parts of broader and more complex behavioural practices, which vary across contexts. For example, smoking behaviour involves lighting the cigarette, bringing it to the mouth, and inhaling the smoke, but this behaviour is embedded within broader practices, such as socialisation, emancipation, or transgression [101,125]. The practice, therefore, is closely linked to the social context in which it develops. However, it is also an expression of the singularity of the subject. Considering the origin of the term style in the artistic field [3], this defines the individual imprint, originality, and creativity in pursuing one’s health practices. Ansbacher [3] offers an interesting reflection on how a child learns to write within a given social and cultural context. The context defines artefacts and writing possibilities, but the child will develop his/her own style, an original modus of handwriting that will differentiate him/her from others. Thus, we can hypothesise that this occurs for individual and collective health practices that develop within a social, cultural, and historical context but reflect an original personal imprint.

The lifestyle concept will be redesigned within a health perspective as a margin of tolerance toward the environment’s infidelities [35]. In this perspective, the healthy subject carries on the subjective experience of exercising their own positioning and can distance themselves from having to adhere uncritically to predefined and a priori rules [35]. At the same time, this subject finds their autonomous capacity for reorganisation and adaptation to the environment within their life project [126]. This definition represents a preliminary proposal for developing a lifestyle model that integrates individual factors with social factors.

This narrative review opens research and intervention questions: how are healthy lifestyles formed and changed in life stages? What meanings, attitudes, and values are central to the development of a healthy lifestyle? It is hypothesised the need to adopt longitudinal studies that integrate qualitative and quantitative methodologies to grasp the various variables underlying health practices, e.g., if and how lifestyles can change under the pressure of collective phenomena such as COVID-19?

Finally, this narrative review has several limitations. First, the in-depth study of the different lifestyle theories appears partial; indeed, the inclusion criteria of the theories considered were related to the most widespread and recognised; thus, not all of them were considered. The proposed definition of lifestyle is currently theoretical, and it is necessary to develop a broader explanatory model to be empirically validated. The risk is to add a new definition without creating an adequate theoretical and research model that serves as a reference for developing health promotion interventions.

## 5. Conclusions

Lifestyle is a complex and multidimensional construct that is still much debated today. Critical health psychology emphasises a vision of a healthy lifestyle not as a set of individual behaviours but as contextually situated and meaningful health practices, the outcome of a complex interactive relationship between the individual and the environment. In this narrative review, we considered the main models and theories on lifestyle in the psychological and sociological fields, analysing them from an internal dimension, external dimension, and temporal dimension. This analysis made it possible to identify the main components that characterise the concept of lifestyle. These components have been included in a new definition of lifestyle in the field of health. A definition of lifestyle was proposed to help clarify the individual and social dimensions and to define several future research directions. Indeed, the construction of a theoretical–explanatory model on the components of lifestyle has made it possible to consider this construct not only as a set of health behaviours but as a system of meanings, attitudes, and values defining health practice models. Considering the practices and antecedent factors to practices may allow for a more systemic approach to research and lifestyle intervention projects. This model also allows for a clearer reference to proceed with a systematic review of the literature on lifestyle studies, defining the inclusion criteria.

## Figures and Tables

**Figure 1 ijerph-20-04427-f001:**
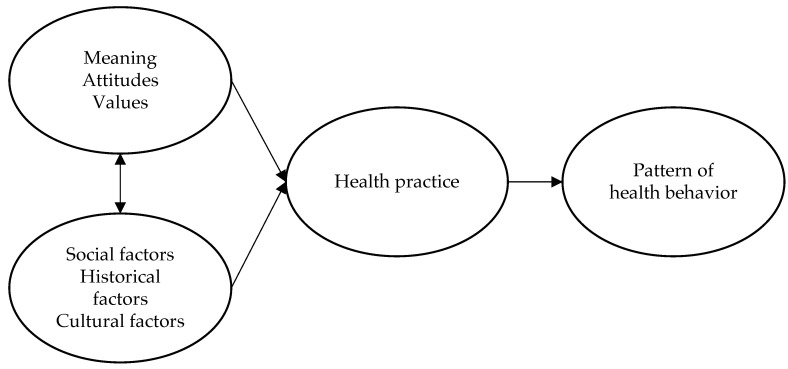
Lifestyle reading model proposed by the new definition.

**Table 1 ijerph-20-04427-t001:** Major explicit definitions of lifestyle in the psychological and sociological literature.

Reference	Definition	Research	Lifestyle Dimension
Adler (1933) [12]	“Their ability to show the individual living, acting, and dying as an indivisible whole in closest context with the tasks of his sphere of life rouses our admiration for their work to the highest degree” […] “the wholeness of his individuality.”	Psychology	Internal, temporal
Allport (1961) [13]	“The complex propriate organisation that determines the ‘total posture’ of a mature life-system.” […] [The lifestyle] “evolves gradually in the course of life, and day by day guides and unifies all, or at least many, of a person’s transactions with life.”	Psychology	Internal, temporal
Coleman (1964) [41]	“The general pattern of assumptions, motives, cognitive styles, and coping techniques that characterise the behavior of a given individual and give it consistency.”	Psychology	Internal, temporal
Schutz et al. (1979) [11]	“The orientation of self, others, and society that each individual develops and follow […] [it] reflects the values and cognitive style of individual. This orientation is derived from personal beliefs based on cultural context and the psycho-social milieu related to the stages of the individual’s life.”	Psychology	Internal
Mitchell, (1983 ) [9]	“We started from the premise that an individual’s array of inner values would create specific matching patterns of outer behavior –that is, of lifestyle.”	Psychology	Internal
WHO (1986) [42]	“Lifestyles are patterns of (behavioural) choices from the alternatives that are available to people according to their socio-economic circumstances and the ease with which they are able to choose certain ones over others.”		
Giddens (1991) [6]	“A lifestyle can be defined as a more or less integrated set of practices which an individual embraces, not only because such practices fulfil utilitarian needs, but because they give material form to a particular narrative of self-identity.” “Lifestyles are routine practices, the routines incorporated into habits of dress, eating, modes of acting and favoured milieus for encountering others; but the routines followed are reflexively open to change in the light of the mobile nature of self-identity.”	Sociology	External, temporal
Veal (1993) [38]	“Lifestyle is the distinctive pattern of personal and social behaviour characteristic of an individual or a group.”	Sociology	External, temporal
Stebbins (1997) [7]	“A lifestyle is a distinctive set of shared patterns of tangible behavior that is organised around a set of coherent interests or social condition or both, that is explained and justified by a set of values, attitudes, and orientations and that, under certain conditions, becomes the basis for a separate, common social identity for its participants” and “lifestyle are not entirely individual […] but are constructed through affiliation and negotiation, by the active integration of the individual and society, which are constantly […] reproduced through each other.”	Sociology	Internal, temporal
Cockerham et al. (1997) [43]	“Collective patterns of health-related behaviour based on choices from options available to people according to their life chances.”	Sociology	External, temporal
Jensen (2009) [44]	“A lifestyle is a pattern of repeated acts that are both dynamic and to some degree hidden to the individual, and they involve the use of artefacts. This lifestyle is founded on beliefs about the world, and its constancy over time is led by intentions to attain goals or sub-goals that are desired. In other words, a lifestyle is a set of habits that are directed by the same main goal.”	Psychology	External, temporal

## Data Availability

No new data were created or analyzed in this study. Data sharing is not applicable to this article.

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
