# Peer review of "Narrative Review and Analysis of the Use of “Lifestyle” in Health Psychology"

_ijerph, 2023, doi:10.3390/ijerph20054427_

Round 1
Reviewer 1 Report
"current" definition. Do you mean the proposed definition or the definition that now exists?
Summarize key points of your findings in the conclusion.
Reviewer 2 Report
The review of how the concept of lifestyle is defined in different traditions is usefull, informative and lucid. But I've had a lot of trouble figuring out where in the research narrative analysis could be linked. That is, the narrative doesn't exist! A narrative analysis suggests a larger historical framework in which the concept has emerged, but I can't find anything similar in the paper other than a few sentences summarizing it. The disposition follows a traditional concept review, dealing with definitions and themes. I advise you to change the title and present the study as a thematic concept analysis.
Reviewer 3 Report
Review for ijerph-2218847
The paper reports a narrative review of the conceptualization of the term “lifestyle” through the theoretical lens of multiple disciplines and its applications in the health psychology field. Although conceptual papers of the term “lifestyle” exist, it could be argued that the present narrative review could contribute to extant knowledge. I feel that this research is ambitious and the paper suffers to some extent from cohesion and coherence issues. Therefore, I believe that significant revisions are required before any further processing.
· The rationale of the importance of researching “lifestyle” is a bit unclear. The authors could write more.
· In the paragraph beginning in line 45, lifestyle is discussed in terms of risk behaviors. However, this paragraph disrupts the flow of the importance of the study. Perhaps, it should come afterwards.
· The introduction would benefit from a clear definition of what is “health” and what is “wellbeing”.
· In line 68, you refer to the field of “health” but it is unclear what field is that. Health may include many disciplines (e.g., psychiatry, physical health) and it is highly ambitious to claim that you will review what “lifestyle” constitutes to different disciplines. Thus, I suggest trying to narrow down the breadth of disciplines you draw upon.
· Given that the review of definitions draw from psychology and sociology, is it accurate to claim that you review the term in relation to the health psychology or “health” field?
· There appears to be a contradiction in lines 73-77, where the author(s) claim to study “lifestyle” through a multidisciplinary perspective and then there is also a focus exclusively through the lens of psychology and sociology.
· Why did the authors select a narrative instead of a systematic review format? This should be clarified given that lifestyle research is abundant.
· If the temporal aspect of lifestyle permeates definitions of psychological (i.e., a personality trait) and sociological perspectives on what is lifestyle, then is it accurate to specify this as a separate aspect/component? One would reasonably expect lifestyle to be something more trait-like and stable.
· For example, the Coleman definition (Table 1) refers to “consistency” of behavioral patterns. This suggests that lifestyle is something stable.
· Line 132: personality style should be changed to personality trait not to be confused with lifestyle (becomes repetitive).
· Line 145: what author?
· Line 156: please rephrase- the meaning is not clear.
· Is “individual psychology” the psychology of individual differences? Please clarify what you mean here.
· It is not clear how needs and need satisfaction is related to lifestyle.
· Lines 188-189: how do the path chosen and the level of satisfaction reflect different lifestyles?
· There is a disproportionate devotion of space to different theoretical perspectives. For instance, Coleman’s and Allport’s definitions had fewer explanations compared to VALS definitions.
· Please refrain from using abbreviations for different perspectives. It becomes confusing trying to keep up with these.
· Line 238: more complex than what?
· It would be helpful to add a paragraph at the end of each section to summarize the commonalities of the definitions of the psychological and sociological perspectives, respectively.
· It would be helpful for the reader to present a graph to outline what factors/components comprise lifestyle, what factors influence lifestyle, and what are the possible outcomes influenced by risky lifestyles.
· In general, the results section reads like a textbook chapter. The definitions are discussed in isolation and there is not a critical evaluation in terms of comparing and contrasting the definitions. This is an important aspect that is missing.
· Greater emphasis should be placed on linking lifestyles with health outcomes.
· On a more practical point of view, I would like to ask what are the practical or research implications of this study?
Round 2
Reviewer 3 Report
The authors have convincingly addressed my concerns. I have no further comments.